# STRuCT-LLM: Unifying Tabular and Graph Reasoning with Reinforcement Learning for Semantic Parsing

## Abstract

Large language models (LLMs) can parse natural language into SQL or Cypher, but remain fragmented—lacking a unified ability to reason across both relational and graph-structured data. We present STRuCT-LLM, a reinforcement learning framework for cross-domain query understanding. Our approach integrates supervised chain-of-thought traces with topology-aware execution rewards, enabling models to acquire complementary reasoning skills: computational and inter-column analysis from SQL and graph traversal from Cypher. On noisy real-world datasets (e.g., SEDE), STRuCT-LLM achieves consistent gains over supervised fine-tuning baselines, including 17% fewer logical errors and 20% fewer data-reference errors, while maintaining robustness under perturbations. Beyond benchmark improvements, we provide a structural analysis of SQL–Cypher equivalence and qualitative case studies showing how unified training resolves errors that single-domain models cannot. These results establish reinforcement learning as a driver of structure-aware generalization across heterogeneous data modalities, paving the way for natural language interfaces to more diverse and unified database systems.

## 1 Introduction

Large language models (LLMs) demonstrate impressive fluency in open-domain generation, yet continue to struggle with *structured reasoning* over tables and graphs (Jiang et al., 2023; Guo et al., 2023). Such reasoning requires grounding entities, composing symbolic constraints, and following precise logical paths—skills central to real-world data systems such as relational databases and knowledge graphs (KGs) (Li et al., 2023; Pourreza & Rafiei, 2023).Executable semantic parsing provides a natural testbed: Text-to-SQL has been widely studied, while Text-to-Cypher remains comparatively underexplored despite being well-suited for graph reasoning (Sui et al., 2024; Ozsoy et al., 2024). Although typically treated in isolation, both tasks demand mapping natural language into compositional, executable programs, and they share core abstractions—schema grounding, filtering, joins, and path composition—making them fertile ground for *cross-formalism transfer*.

Despite rapid progress in LLM reasoning, current approaches remain siloed: SQL benchmarks capture relational computations, while Cypher benchmarks capture graph traversals. No framework has unified these domains, leaving LLMs unable to reason seamlessly across heterogeneous structured data (Galkin & Doguparty, 2025; Zhang et al., 2024). We argue that SQL and Cypher are not merely separate benchmarks, but *complementary, executable training signals*: SQL imparts relational operations and inter-column computation, while Cypher imparts multi-hop traversal and neighborhood aggregation. Bridging them unlocks reasoning skills that neither modality alone can teach, opening the door to more capable database interfaces, richer knowledge graph exploration, and genuinely unified reasoning systems.

To address this gap, we propose **STRuCT-LLM**, a unified reinforcement learning framework that combines Chain-of-Thought (CoT) supervision with Group Relative Policy Optimization (GRPO) (Shao et al., 2024). We prompt instruction-tuned LLMs (Qwen2.5-1.5B, 14B, QwQ-32B, Qwen3-14B) (Yang et al., 2024a) with decomposed reasoning traces and optimize them with execution-, structural-, and syntax-based rewards. For Cypher, we introduce a *topology-aware reward* that evaluates correctness over nodes, edges, and connectivity—providing continuous, interpretable feedback that

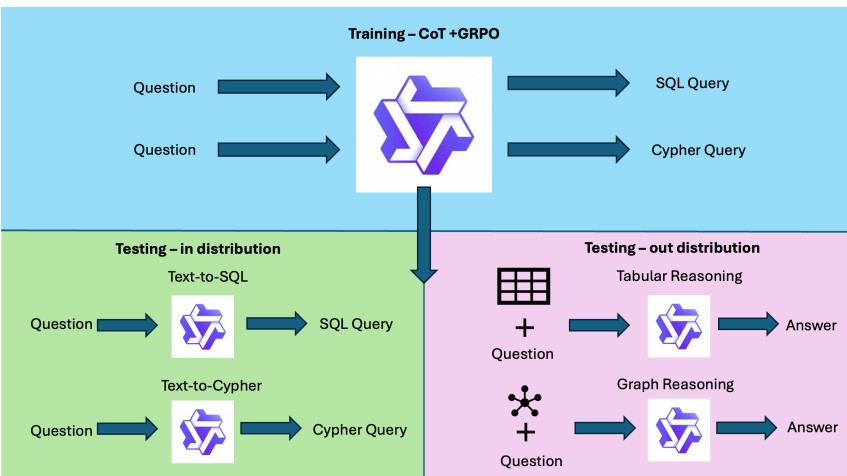

Figure 1: **Overview of our unified training and evaluation setup.** Joint training on Text-to-SQL and Text-to-Cypher using Chain-of-Thought supervision and RL (Task 1). Evaluation covers generalization to SQL/Cypher (Task 2), zero-shot transfer to tabular (Task 3), and graph QA (Task 4), testing for transferable reasoning from executable semantic parsing.

addresses a longstanding bottleneck in graph semantic parsing. Our joint training regime interleaves SQL and Cypher across both supervised and RL stages, encouraging structural transfer while reducing task-specific overfitting. This builds on the recent momentum around grounding LLM training in verifiable and interpretable feedback (Pourreza et al., 2025; Stoisser et al., 2025; Yuan et al., 2024), and extends it by demonstrating that SQL–Cypher synergy yields measurable gains in noisy and unseen-schema settings.

Empirically, STRuCT-LLM yields consistent gains on Spider, SEDE, EHRSQL, and BIRD benchmarks, and improves robustness under noisy and perturbed settingsHazoom et al. (2021). It reduces logical and data-reference errors by 17–20% compared to single-domain baselines, while maintaining strong execution accuracy. Beyond parsing, our models exhibit zero-shot transfer to table and KG QA (CRT-QA, TableBench, CR-LT-KGQA) without direct QA supervision, and even generalize to unseen query languages such as MQL. These findings reinforce our central claim: executable queries are not only evaluation targets, but also *vehicles for teaching LLMs structured reasoning*.

CONTRIBUTIONS

This paper makes the following contributions:

- **Unified SQL–Cypher training.** We introduce the first reinforcement learning framework that jointly optimizes semantic parsing across relational (SQL) and graph-based (Cypher) queries, demonstrating complementary reasoning skills and cross-formalism transfer.
- **Topology-aware reward.** We design a subgraph-based reward that evaluates node, edge, and connectivity correctness, enabling fine-grained, structure-sensitive optimization for Cypher parsing and moving beyond binary execution feedback.
- **Robust generalization.** STRuCT-LLM consistently outperforms single-task baselines across multiple datasets (Spider, SEDE, EHRSQL, BIRD), shows robustness to noisy/perturbed queries, and transfers zero-shot to unseen query languages (MQL) and QA tasks (CRT-QA, TableBench, CR-LT-KGQA).

## 2 RELATED WORK

**Semantic parsing over structured data.** Executable semantic parsing provides a supervised testbed for structured reasoning in LLMs. In the SQL domain, benchmarks such as Spider Yu et al. (2018), BIRD Li et al. (2023), and WikiSQL enabled systematic study of Text-to-SQL, with advances

in schema linking Liu et al. (2024); Katsogiannis-Meimarakis & Koutrika (2023), execution-guided decoding Huang et al. (2024), and structured intermediate representations Rubin & Berant (2021). Reinforcement learning has also been applied, optimizing for execution correctness Yang et al. (2024b), partial rewards tailored to SQL Pourreza et al. (2025), or robustness in noisy queries Stoisser et al. (2025). Yet this literature treats SQL in isolation, leaving open the question of whether relational supervision can transfer to other structured modalities.

**Parsing graph queries.** Cypher parsing introduces additional complexity due to dynamic topologies and path-based reasoning. Progress has been constrained by data scarcity Zou et al. (2024), but recent efforts include MedT2C Zhong et al. (2025), SynthCypher Tiwari et al. (2025), and large-scale datasets with fine-tuned baselines Ozsoy et al. (2024); Ozsoy (2025). These works highlight structural parallels between relational and graph schemas, but rely almost exclusively on supervised fine-tuning, without exploring reinforcement signals aligned with graph connectivity.

**Cross-formalism and unified approaches.** Multi-formalism parsing has been explored through synthetic datasets such as SM3-Text-to-Query Sivasubramaniam et al. (2024), spanning SQL, SPARQL, Cypher, and MQL. Unified frameworks like UnifiedSKG Xie et al. (2022) and multi-task adapters demonstrate benefits of heterogeneous supervision. Similar ideas have extended to SPARQL Yin et al. (2021); Xu et al. (2023) and MQL Lu et al. (2025), showing that structural bias can promote generalization across symbolic languages. In downstream evaluation, benchmarks such as CRT-QA Zhang et al. (2023), TableBench Wu et al. (2025), and CR-LT KGQA Guo et al. (2024) emphasize robustness, commonsense reasoning, and rare entities—settings where structure-aware models are especially valuable.

**Our contribution.** Prior work has studied SQL-only parsing, Cypher-specific corpora, or synthetic multi-formalism settings, but none provide a reinforcement learning framework that jointly optimizes across relational and graph-structured formalisms. Our work closes this gap with three innovations: (i) integrating chain-of-thought supervision with RL to encourage compositional reasoning, (ii) introducing a topology-aware reward for Cypher that evaluates node, edge, and subgraph correctness, and (iii) showing that SQL and Cypher are not only individually tractable but *mutually beneficial*. Relational joins strengthen Cypher's ability to handle computation, while graph traversal improves SQL schema grounding. This synergy yields consistent improvements on in-distribution parsing and noisy real-world benchmarks, and transfers zero-shot to unseen query formalisms and downstream QA tasks—evidence that executable queries can serve as a universal training signal for structure-aware reasoning.

## 3 METHODOLOGY

### 3.1 REPRESENTATIONAL EQUIVALENCE OF RELATIONAL TABLES AND PROPERTY GRAPHS

We treat relational tables and property graphs as two views of the same data. Under mild, standard assumptions (declared primary keys and typed properties), there are lossless mappings between any finite relational database and a property graph. Concretely: (i) each row identified by its key becomes a node; (ii) non-key attributes become node properties or value nodes connected by typed edges; and (iii) foreign keys become edges between key nodes. Conversely, any property graph can be materialized in relational form using canonical node, edge, node-property, and edge-property tables (see Appendix A for proofs). This equivalence implies that SQL and Cypher queries, though syntactically distinct, both reduce to chains of filters, joins/path constraints, and projections over the same underlying space. It motivates joint training: exposing a model to both formalisms expands coverage of this shared space and promotes transfer of structural priors (e.g., key–foreign-key joins $\longleftrightarrow$ path patterns). It also grounds our reward design: SQL benefits from component-level matching, while Cypher benefits from topology-aware feedback on subgraph structure.

### 3.2 PROBLEM FORMULATION

Let $\mathcal{Q}$ denote the query space generated by applying a sequence of operations $\mathcal{O}$ (selection, projection, join, pattern match, etc.) to an initial dataset $D$ with schema $S$. Each query $Q \in \mathcal{Q}$ is a composite

function

$$Q(D) = \mathcal{O}_n(\mathcal{O}_{n-1}(\cdots \mathcal{O}_1(D) \cdots)),$$

producing a valid result $D' \subseteq \mathcal{R}_{\text{Result}}$. Semantic parsing is the task of mapping a natural language input $X$ to a target query $Q$ in $\mathcal{Q}$ that retrieves the correct result from $D$. Building on the shared space of Section 3.1, we train models jointly on Text-to-SQL and Text-to-Cypher, with two stages: (1) supervised fine-tuning on synthetic Chain-of-Thought (CoT) traces, and (2) reinforcement learning with execution-, syntax-, and structure-aware rewards Guo et al. (2025).

## 3.3 DATASETS

We construct training corpora for both stages. For supervised fine-tuning, we generate CoT traces for 24 Text-to-SQL datasets[1] and the Neo4j Text-to-Cypher dataset Ozsoy et al. (2024) using a structured prompting pipeline inspired by Boubnovski et al. (2025). Each reasoning trace is produced and verified by LLMs for coherence (details in Appendix G). For reinforcement learning, we sample 3.5k queries from BIRD Li et al. (2023) (SQL) and 3.5k from Neo4j Text-to-Cypher Ozsoy et al. (2024), ensuring non-overlap with fine-tuning data. Single-task baselines use 7k examples, while joint training balances SQL and Cypher to encourage fair cross-formalism transfer.

## 3.4 REINFORCEMENT LEARNING FRAMEWORK

We adopt Group Relative Policy Optimization (GRPO) (Shao et al., 2024), an extension of PPO that compares multiple candidate outputs for the same input and optimizes their relative advantages. This design provides more nuanced feedback for complex reasoning tasks than single-output RL. For completeness, the full optimization objective is included in Appendix C.

## 3.5 REWARD DESIGN

Binary execution accuracy is insufficient for nearly correct queries, as it lacks gradient information. We therefore design continuous, language-specific rewards:

1. **LLM Judge Reward:** Uses o3-mini Jaech et al. (2024) as a classifier of overall query quality, following Xin et al. (2024).

2. **String Matching Reward:** Computes the longest matching subsequence between predicted and gold queries, encouraging syntactic fidelity.

3. **Structural Consistency Rewards:** Tailored to each query language:
   - **SQL:** Component-Level Matching (F1 over SELECT, WHERE, GROUP BY, etc.) as in Stoisser et al. (2025); Nguyen et al. (2025).
   - **Cypher:** *Graph Edit Distance Reward*, our novel contribution. Predicted and gold Cypher queries are converted into property graphs; the reward is normalized by graph size:
$$R_{\text{GED}} = 1 - \frac{\text{GED}}{\max(\text{size}_1, \text{size}_2)}.$$
     This encourages models to recover correct topology even when surface forms differ.

Together, these rewards provide complementary feedback: global quality (judge), surface similarity (string), and structural fidelity (components/graph).

## 3.6 JOINT TRAINING

We interleave SQL and Cypher queries during RL, using task-specific reward combinations:

$$R_{\text{SQL}} = R_{\text{judge}} + R_{\text{string}} + R_{\text{component}}, \quad R_{\text{Cypher}} = R_{\text{judge}} + R_{\text{string}} + R_{\text{GED}}.$$

Weights are kept equal ($w_1 = w_2 = w_3 = 1$) for simplicity. This setup encourages bidirectional transfer: SQL contributes computational and inter-column reasoning, while Cypher contributes traversal and topology awareness. We find this balance is key to structure-aware generalization across formalisms.

---

[1]https://huggingface.co/datasets/Clinton/Text-to-sql-v1

## 4 EXPERIMENTS

### 4.1 DATASET

We evaluate our model based on two primary criteria: (1) its performance on semantic parsing tasks, specifically through its generalization to novel databases, is assessed using Spider Yu et al. (2018), EHRSQL Lee et al. (2022) and BIRD minidev [2] for Text-to-SQL and Text2Cypher Ozsoy et al. (2024) for Text-to-Cypher; and (2) its transfer capability to out-of-distribution tasks in the Table/KG Question Answering, which reflects its induced reasoning capabilities, is evaluated using CRT-QA Zhang et al. (2023) for tabular and CR-LT KGQA Guo et al. (2024) for KG QA. Additionally, we leverage the SM3 dataset Sivasubramaniam et al. (2024), which provides semantically equivalent queries across SQL, Cypher, and MQL, enabling direct comparison of model performance across different query languages. Further details can be found in Appendix B.

### 4.2 EXPERIMENTAL SETUP

In our experiments, we use both non-reasoning models (Llama3-8B, Qwen2.5-1.5B, Qwen2.5-14B) and reasoning models (QwQ-32B, Qwen3-14B) as baselines. We utilize VERL[3] for training the 14B and 32B models. To enhance efficiency, Unsloth[4] is employed, limiting the training to QLora Adapters with a LoRA rank of 16 for all 1.5B models. For Supervised Fine-Tuning, we adopt a learning rate of $1e^{-6}$ for larger models (14B and 32B) and $1e^{-4}$ for smaller models. The batch size is set to 512 for big models and 8 for small models. For GRPO, consistent with Pourreza et al. (2025), a constant learning rate scheduler is applied. Training is conducted for one epoch with a learning rate of $1e^{-6}$ for large models and $1e^{-5}$ for small models. Additional hyperparameters for GRPO include a number of generations $G = 6$, with batch sizes of 1024 for large models and 32 for small models, using the `PagedAdamW8bit` optimizer. For robust evaluation, we sample 200 datapoints from each dataset five times and report mean scores and standard deviations. For training any size of gpu can be used for 1.5B models, we use 1 A100 GPU for 8B and 14B (10 h and 20 h) models, and 32 H100 GPUs for 20 h for the 32B model. Additional hyperparameters are available in our GitHub repository.

### 4.3 IN-CONTEXT PERFORMANCE

Table 1 presents the performance on semantic parsing tasks. Training on a single executable language (SQL or Cypher) induces *slight positive transfer* to the other: for instance, training Qwen2.5-14B-Instruct on Cypher alone improves its SQL performance on Spider (EXE: 71.7 vs. 69.8), and vice versa for Text2Cypher. This cross-task improvement suggests structural similarities—such as schema grounding and compositional filtering—shared between the two tasks.

Motivated by this, we train models on both SQL and Cypher jointly. The resulting models demonstrate strong *synergy*, consistently outperforming single-task variants across all base models. For example, Qwen2.5-14B-trained-Both outperforms both SQL-only and Cypher-only versions on all tasks.

Notably, our QwQ-32B-trained-Both model *approaches or exceeds* the performance of proprietary instruction-tuned models such as o3 and o3 mini, particularly on Text2Cypher. This suggests that dual-format supervision is a powerful training signal, enabling open models to compete with the best closed models on structural parsing tasks.

These results are on main benchmarks within their respective domains; however, they are not comparable to each other directly. Hence, in Appendix F, we include additional results on the SM3-Text-to-Query Dataset Sivasubramaniam et al. (2024), which provides semantically equivalent queries across SQL, Cypher, and MongoDB Query Language (MQL). This dataset allows us to evaluate how well our training approach generalizes to unseen query languages.

We observe that base models perform better on SQL compared to Cypher and show significantly weaker performance on MQL, reflecting the data availability during pretraining (SQL being the most abundant and MQL the scarcest). On the other hand, our models achieve superior performance in Cypher compared to SQL, potentially due to Cypher's relative simplicity or the greater effectiveness

---

[2] https://GitHub.com/bird-bench/mini_dev
[3] https://GitHub.com/volcengine/verl
[4] https://GitHub.com/unslothai/unsloth

Table 1: **In-context learning performance on Text-to-SQL (Spider, EHRSQL, BIRD minidev with evidence) and Text-to-Cypher (Text2Cypher) tasks**. Metrics include exact match (EM), execution accuracy (EXE), BLEU score and execution-based F1 score ($F1_{exe}$, per Lee et al. (2022) ). Models are labeled as 'trained-SQL', 'trained-Cypher', or 'trained-Both' based on fine-tuning. Top two results per dataset are bolded.

| Model | Text2Cypher | | BIRD | Spider | EHRSQL |
|---|---|---|---|---|---|
| | EM | BLEU | EXE | EXE | $F1_{exe}$ |
| o3 | 4.0 ±0.6 | 25.0 ±2.2 | 54.0 ±4.5 | **76.8** ±1.6 | **46.5** ±2.5 |
| o3-mini | 4.5 ±1.1 | 25.4 ±2.2 | 50.0 ±2.1 | 74.5 ±3.2 | 35.5 ±1.5 |
| Llama3-8B-instruct | 3.2 ±1.1 | 16.7 ±1.4 | 39.0 ±3.1 | 62.8 ±3.2 | 23.4 ±3.8 |
| Llama3-8B-trained-SQL | 4.1 ±1.0 | 18.9 ±1.5 | 45.3 ±3.2 | 65.5 ±2.6 | 29.1 ±3.2 |
| Llama3-8B-trained-Cypher | 6.1 ±1.3 | 22.5 ±1.7 | 42.1 ±3.3 | 64.5 ±2.8 | 27.1 ±3.2 |
| Llama3-8B-trained-Both | 6.6 ±1.2 | 22.7 ±1.8 | 45.4 ±2.9 | 67.1 ±2.3 | 30.2 ±3.4 |
| Qwen2.5-14B-instruct | 3.5 ±1.2 | 18.5 ±1.6 | 43.3 ±3.4 | 69.8 ±3.6 | 26.0 ±4.2 |
| Qwen2.5-14B-trained-SQL | 4.5 ±1.1 | 21.0 ±1.7 | 50.3 ±3.5 | 72.8 ±2.9 | 32.3 ±3.6 |
| Qwen2.5-14B-trained-Cypher | 6.8 ±1.4 | 25.0 ±1.9 | 46.8 ±3.7 | 71.7 ±3.1 | 30.1 ±3.5 |
| Qwen2.5-14B-trained-Both | **7.3** ±1.3 | 25.2 ±2.0 | 50.4 ±3.2 | 74.5 ±2.6 | 33.6 ±3.8 |
| QwQ-32B | 3.7 ±1.1 | 19.3 ±2.3 | 50.0 ±3.0 | 69.8 ±3.6 | 35.0 ±2.6 |
| QwQ-32B-trained-Both | **12.0** ±1.5 | **33.4** ±3.6 | **55.3** ±3.0 | **79.2** ±4.0 | 43.0 ±1.6 |

| Model | Spider | | SEDE | |
|---|---|---|---|---|
| | Logical Err ↓ | Ref Err ↓ | Logical Err ↓ | Ref Err ↓ |
| SQL-only | 23.5% | 18.7% | 25.1% | 19.4% |
| Cypher-only | 24.1% | 20.3% | 26.2% | 20.1% |
| **Struct-LLM (Joint)** | **19.5%** | **16.2%** | **20.7%** | **15.8%** |

Table 2: Error-type analysis on **Spider** and **SEDE**. **Struct-LLM** reduces logical errors by ∼17% and data-reference errors by ∼20% compared to single-domain baselines, while also improving execution accuracy

of our structural reward mechanism in tackling graph-based tasks. Additional results on Qwen3-14B are in Appendix D.

**Error Breakdown.** Beyond execution accuracy, we examine the nature of errors on Spider and SEDE by categorizing failures into two groups: (i) *logical errors*, where the overall computation or operator choice is incorrect (e.g., wrong aggregation, join condition, or traversal path), and (ii) *data-reference errors*, where the query structure is correct but the model references the wrong schema element (e.g., wrong column, table, or entity). As shown in Table 2, Struct-LLM reduces logical errors by approximately 17% and data-reference errors by approximately 20% compared to SQL-only and Cypher-only baselines. This indicates that the benefits of joint training extend beyond raw accuracy: the model makes qualitatively better decisions about operators and schema elements. These reductions are consistent with our qualitative case studies (Section 5), where SQL-to-Cypher transfer improved computational reasoning (e.g., percentiles), and Cypher-to-SQL transfer improved traversal reasoning (e.g., distinct coauthor counting).

## 4.4 OUT-OF-CONTEXT PERFORMANCE

Table 3 presents the performance metrics on structural understanding tasks. Training on executable semantic parsing improves LLMs' ability to understand structured data more broadly, yielding better zero-shot performance on unseen question answering tasks. For example, both SQL- and Cypher-trained models outperform baselines on CRT-QA and CR-LT-KGQA, showing that execution-based reward helps models learn to reason over structured representations.

Moreover, joint training again provides strong synergy: Qwen2.5-14B-trained-Both, for example, achieves the best performance across Qwen2.5-14B models, with 62.5 EM on CRT-QA and 86.3

Table 3: **Out-of-context performance on structural question answering**. Results are reported on tabular (CRT-QA, TableBench) and knowledge graph QA (CR-LT-KGQA). Metrics include exact match (EM) and ROUGE. Models are labeled as 'trained-SQL', 'trained-Cypher', or 'trained-Both' based on fine-tuning. Top two results per dataset are bolded.

| Model | CRT-QA | TableBench | | CR-LT KGQA |
|---|---|---|---|---|
| | EM | EM | Rouge | EM |
| o3 | **61.5** ±2.3 | **68.0** ±4.4 | **73.1** ±3.5 | **92.0** ±0.7 |
| o3-mini | 45.2 ±4.0 | 64.8 ±4.4 | 68.6 ±3.8 | **91.7** ±2.3 |
| Llama3-8B-instruct | 50.0 ±1.2 | 50.0 ±4.2 | 52.5 ±3.7 | 72.2 ±1.1 |
| Llama3-8B-trained-SQL | 53.9 ±1.4 | 56.6 ±3.5 | 55.8 ±3.2 | 74.3 ±1.2 |
| Llama3-8B-trained-Cypher | 51.0 ±1.7 | 53.6 ±3.9 | 54.9 ±3.5 | 77.1 ±1.1 |
| Llama3-8B-trained-Both | 56.3 ±1.5 | 57.4 ±3.2 | 59.0 ±3.5 | 77.7 ±1.3 |
| Qwen2.5-14B-instruct | 55.5 ±1.3 | 55.5 ±4.7 | 58.3 ±4.1 | 80.2 ±1.2 |
| Qwen2.5-14B-trained-SQL | 59.9 ±1.6 | 62.9 ±3.9 | 62.0 ±3.5 | 82.5 ±1.3 |
| Qwen2.5-14B-trained-Cypher | 56.7 ±1.9 | 59.5 ±4.3 | 61.0 ±3.9 | 85.7 ±1.2 |
| Qwen2.5-14B-trained-Both | **62.5** ±1.7 | 63.8 ±3.5 | 65.5 ±3.9 | 86.3 ±1.4 |
| QwQ-32B | 45.7 ±4.3 | 63.3 ±5.6 | 67.2 ±4.9 | 89.8 ±1.4 |
| QwQ-32B-trained-Both | 57.0 ±2.8 | **68.7** ±3.6 | **73.4** ±2.7 | 91.3 ±1.7 |

accuracy on CR-LT-KGQA—gains of 2.6 and 0.8 points over the best single-task variants. These results affirm that multi-format semantic supervision promotes generalization across structured modalities.

Additionally, the results in Appendix F show that our models improve over baselines in few-shot settings and generalize effectively to the completely unseen structural language MQL. These findings reinforce the broad applicability of our approach to diverse structured reasoning tasks.

## 4.5 ABLATION STUDIES

We conducted comprehensive ablation studies to evaluate: (1) the impact of different training configurations (SQL-only, Cypher-only, and joint training), and (2) the contribution of each reward component in our GRPO framework. Our analysis employed Qwen2.5-1.5B with results presented in Tables E1, E2.

**Training Regime Ablation (SFT vs. SFT+GRPO)** To isolate the effect of GRPO beyond supervised fine-tuning, we include a training-regime ablation (Base, SFT-only, SFT+GRPO); see Table E1. SFT+GRPO consistently outperforms SFT-only across SQL-only, Cypher-only, and joint training. Gains are largest on structure-heavy tasks (e.g., Spider and KGQA), underscoring the need for GRPO on top of SFT.

**Reward Component Ablation** We evaluate the impact of different reward components (string matching, LLM-based, and structural) through systematic ablation, denoted by 1/0 in model names. The full reward configuration (111) consistently achieves optimal performance, highlighting the complementary nature of these components. Ablation experiments reveal the distinct role of each component: removing string matching rewards (011) causes notable performance drops, especially in semantic parsing tasks, while disabling LLM-based rewards (101) leads to moderate degradation. The structural reward proves particularly crucial—its removal (110) significantly impacts performance, especially on QA tasks where structural comprehension is essential.

Overall, these analyses demonstrate the synergistic benefits of our multi-component reward structure and our training approach. Additional ablation details are provided in Appendix E2.

## 4.6 QUALITATIVE CASE STUDIES: BIDIRECTIONAL SQL–CYPHER TRANSFER

While aggregate benchmarks establish overall improvements, we further analyze concrete queries where single-domain models fail but STRuCT-LLM succeeds. Figure 2 illustrates two representative

### Case A — SQL → Cypher: Distributional computation (median & IQR)

*NLQ: "For each department, return the median salary and the IQR of salaries."*

**SQL-only ✓ correct (ordered-set aggregates)**

```
SELECT d.name AS dept,
       PERCENTILE_CONT(0.5)
         WITHIN GROUP (ORDER
↪ BY e.salary) AS median,
       PERCENTILE_CONT(0.75)
         WITHIN GROUP (ORDER
↪ BY e.salary)
     - PERCENTILE_CONT(0.25)
         WITHIN GROUP (ORDER
↪ BY e.salary) AS iqr
FROM Employee e
JOIN Department d ON d.id = e.
↪ dept_id
GROUP BY d.name;
```

**Cypher-only ✗ wrong (mean/stDev used instead of median/IQR)**

```
MATCH (d:Department)<-[:
↪ BELONGS_TO]-(e:Employee)
RETURN d.name AS dept,
       avg(e.salary)    AS
↪ median,   // wrong: mean
↪ != median
       stDev(e.salary) AS iqr;
↪        // wrong: stDev !=
↪ IQR
```

**Joint (Cypher) ✓ correct via list-based percentiles**

```
MATCH (d:Department)<-[:
↪ BELONGS_TO]-(e:Employee)
WITH d, collect(e.salary) AS
↪ sals
WITH d, apoc.coll.sort(sals)
↪ AS xs
WITH d, xs,
     toInteger(size(xs)*0.25)
↪ AS i1,
     toInteger(size(xs)*0.50)
↪ AS i2,
     toInteger(size(xs)*0.75)
↪ AS i3
RETURN d.name AS dept,
       xs[i2]        AS median,
↪
       xs[i3]-xs[i1] AS iqr;
```

### Case B — Cypher → SQL: Multi-hop traversal (distinct coauthors)

*NLQ: "List authors who have collaborated with at least two other authors on the same paper."*

**SQL-only ✗ wrong (double-counts, missing DISTINCT)**

```
SELECT a1.name
FROM Author a1
JOIN Author a2 ON a1.paper_id
↪ = a2.paper_id
WHERE a1.id <> a2.id
GROUP BY a1.name
HAVING COUNT(a2.id) >= 2;
↪ -- wrong: should be COUNT(
↪ DISTINCT a2.id)
```

**Cypher-only ✓ correct (natural traversal + DISTINCT)**

```
MATCH (a1:Author)-[:WROTE]->(
↪ p:Paper)<-[:WROTE]-(a2:
↪ Author)
WHERE a1 <> a2
WITH a1, COUNT(DISTINCT a2)
↪ AS coauthors
WHERE coauthors >= 2
RETURN a1.name;
```

**Joint (SQL) ✓ corrected aggregation**

```
SELECT a1.name
FROM Author a1
JOIN Paper p ON a1.id = p.
↪ author_id
JOIN Author a2 ON a2.id = p.
↪ author_id
 AND a1.id <> a2.id
GROUP BY a1.name
HAVING COUNT(DISTINCT a2.id)
↪ >= 2;
```

Figure 2: **Bidirectional transfer between SQL and Cypher with the NLQ outside the code.** *Case A* (SQL→Cypher): SQL's ordered-set computations (median/IQR) transfer to graphs, where correct Cypher requires list collection, sorting, and percentile indexing. *Case B* (Cypher→SQL): Graph traversal patterns transfer back to SQL, correcting aggregation logic for distinct coauthor counting.

cases of bidirectional transfer. **Case A (SQL → Cypher).** The NLQ asks: "For each department, return the median salary and the interquartile range (IQR) of salaries." SQL expresses this naturally via ordered-set aggregates (PERCENTILE). In contrast, Cypher-only models misinterpret the query, substituting mean and standard deviation for median and IQR. STRuCT-LLM, however, learns to adapt the SQL-style computation into Cypher by collecting salaries, sorting them, and indexing into the list—thereby producing the correct graph query. This demonstrates that computational and inter-column reasoning transfers from SQL into Cypher. **Case B (Cypher → SQL).** The NLQ asks: "List authors who have collaborated with at least two other authors on the same paper." Graph models capture this easily through multi-hop traversal and distinct coauthor counting. SQL-only models, however, double-count coauthors due to missing DISTINCT. STRuCT-LLM corrects this by transferring Cypher's traversal pattern into SQL, yielding the correct aggregation. This shows that multi-hop relational reasoning transfers from Cypher into SQL. Together, these examples highlight how STRuCT-LLM not only reduces surface-level logical errors but also enables structure-aware generalization: SQL imparts computational analysis skills to Cypher, while Cypher imparts traversal skills to SQL. This bidirectional complementarity underlies the robustness gains observed across datasets.

## 5 DISCUSSION

Our findings provide strong evidence that structured reasoning across distinct data modalities—relational tables and knowledge graphs—can be unified through a shared training paradigm. By combining Text-to-SQL and Text-to-Cypher supervision with reinforcement learning, STRuCT-LLM acquires inductive biases that transfer across formalisms, yielding consistent accuracy gains and qualitatively fewer reasoning errors.

A key insight is the observed *cross-formalism transfer*: SQL supervision improved Cypher's ability to compute numerical statistics (e.g., percentiles, IQR), while Cypher supervision enhanced SQL's handling of multi-hop traversal and distinct counting. Despite non-overlapping schemas, the model internalized abstractions such as joins, filters, and paths, suggesting an emerging capacity for schema-agnostic reasoning.

Our topology-aware reward for Cypher, built on graph edit distance and subgraph matching, was critical to these gains. Unlike binary execution signals, this continuous, structure-sensitive reward provided fine-grained optimization and interpretable feedback, a contribution that extends beyond this specific setup to broader semantic parsing research.

Finally, joint training acted as a natural regularizer, reducing overfitting to dataset-specific patterns. This improved robustness is reflected not only in benchmark scores but also in *zero-shot transfer* to downstream QA tasks (CRT-QA, CR-LT-KGQA), where STRuCT-LLM delivered non-trivial improvements without direct supervision. These results reinforce our central claim: executable queries serve as a powerful scaffold for teaching LLMs structured reasoning.

## 6 LIMITATIONS

While STRuCT-LLM demonstrates robust gains, several limitations remain. First, although our topology-aware reward improves structural fidelity, it does not yet account for query efficiency. Second, our evaluation assumes schema-aware inputs; extensions to interactive or schema-free parsing remain unexplored. Finally, our two-stage training approach, combining supervised fine-tuning with reinforcement learning, is more resource-intensive than single-stage baselines. For the largest 32B model, training on 32 H100 GPUs required roughly 20 hours in total. Smaller variants (1.5B, 8B, 14B) train much faster under the smaller setup using A100s, and we provide all configurations and hyperparameters to ensure reproducibility. While computationally significant at the high end, we believe this cost is justified by the strong generalization and robustness of STRuCT-LLM.

We mitigated overlap between training and evaluation corpora, but cannot fully rule out indirect leakage (e.g., Wikipedia tables appearing in Spider). Similarly, some QA benchmarks may share entities with parsing datasets, though they target distinct reasoning skills.

Addressing these limitations—through more diverse datasets, efficiency-aware objectives, and schema-free interactive settings—represents important future work.

## 7 CONCLUSION

We introduced **STRuCT-LLM**, a unified semantic parsing framework that combines chain-of-thought supervision with reinforcement learning across SQL and Cypher. Our approach consistently improves performance on Spider, SEDE, EHRSQL, and BIRD, and reduces systematic errors ∼17% fewer logical and ∼20% fewer data-reference mistakes relative to single-domain baselines.

A key contribution is our topology-aware reward for Cypher, which provides continuous,structure-sensitive feedback and enables cross-formalism transfer: SQL supervision improves Cypher's handling of computations, while Cypher supervision enhances SQL's multi-hop reasoning. Beyond parsing, STRuCT-LLM demonstrates zero-shot gains on QA tasks, confirming that executable queries serve as a robust scaffold for structured reasoning.

Future directions include extending the training to additional query languages (e.g., SPARQL, MQL), schema-free and interactive parsing, and deeper probes of compositional generalization.

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

## A   TABLE–GRAPH CONVERSIONS ADN EQUIVALENCE

The following conversion algorithms together establish the equivalence of table and graph spaces.

**Graph to Table Conversion Algorithm**    One table specifies edges (e.g., source, relationship, target), another captures node attributes, and a third describes edge attributes.

**Table to Graph Conversion Algorithm**    We define a node type `primary_key` and an additional node type for each table in the schema. Each value in the `primary_key` column of any table is converted into a node of type `primary_key`. Similarly, every other entry in the tables is converted into a node corresponding to the table it belongs to (e.g., nodes from the table called `Age` will have the type `Age`). We then connect each `primary_key` node to all the other nodes in the same row as the primary key using edges, specifying the column name as the edge type.

### A.1   WORKED EXAMPLE

Take the following toy database:

**Age Table**

| patient_name | age | primary_key |
|---|---|---|
| Anna | 30 | 1 |
| Phil | 55 | 2 |

**Height Table**

| patient_id | height | primary_key |
|---|---|---|
| 1 | 165 | 3 |
| 2 | 180 | 2 |

with the following schema:

```
CREATE TABLE Age (
  patient_name VARCHAR(255),
  age INT,
  primary_key INT PRIMARY KEY
);

CREATE TABLE Height (
  patient_id INT,
  height INT,
  primary_key INT PRIMARY KEY
);
```

Our transformation would result in this graph:

**Primary Key: 1 Primary Key:** `primary_key:   1`
   **Connected to:**    • Patient Name: Anna (patient_name attribute) via 'Age' relationship
        • Age: 30 (age attribute) via 'Age' relationship

**Primary Key: 2 Primary Key:** `primary_key:   2`
   **Connected to:**    • Patient Name: Phil (patient_name attribute) via 'Age' relationship
        • Age: 55 (age attribute) via 'Age' relationship
        • Patient ID: 2 (patient_id attribute) via 'Height' relationship
        • Height: 180 (height attribute) via 'Height' relationship

**Primary Key: 3 Primary Key:** `primary_key:   3`

**Connected to:**   • Patient ID: 1 (patient_id attribute) via 'Height' relationship
• Height: 165 (height attribute) via 'Height' relationship

Note that the relationship types here

```
(:PrimaryKey)-[:age]->(:Age)
(:PrimaryKey)-[:height]->(:Height)
(:PrimaryKey)-[:patient_id]->(:Height)
(:PrimaryKey)-[:patient_name]->(:Age)
```

correspond exactly to the columns we can see in the table schema above.

The SQL query:

```
SELECT height FROM Height;
```

on the database now corresponds to the Cypher query:

```
MATCH (v)<-[:height]-(:primary_key)
RETURN v;
```

## B  DATASETS

As detailed in Table B1, we utilize a variety of datasets for SQL and Cypher.

### B.1  SFT TRAINING

**SQL Dataset:** We utilize a collection of 24 publicly available Text-to-SQL datasets, selected for their moderate complexity to facilitate reliable chain-of-thought (CoT) trace generation. This collection is part of a dataset available on Hugging Face[5], which is a large-scale compilation of natural language to SQL examples spanning various domains. This benchmark includes 26 individual datasets that cover academic records, medical databases, entertainment metadata, government statistics, and more. Notable examples include WikiSQL Zhong et al. (2017), ATIS Hemphill et al. (1990), Criteria2SQL Fang et al. (2022), SEDE Hazoom et al. (2021), SQuALL Shi et al. (2020), and NVBench Wang & Crespo-Quinones (2023), along with public domain table corpora such as IMDb, Yelp, and datasets on historical sports or wildfires. We exclude Spider Yu et al. (2018) and EHRSQL Lee et al. (2022) from our training process, as they are designated for evaluation purposes, thus resulting in a total of 24 datasets used for training.

**Cypher Dataset:** The Text-to-Cypher dataset is derived from Ozsoy et al. (2024), encompassing 16 standardized public datasets that represent a broad array of graph query scenarios.

This strategic selection ensures comprehensive foundational development in both SQL and Cypher querying languages, facilitating an effective cold start for subsequent GRPO training.

### B.2  EVALUATION

We evaluate the model on two main criteria. First, semantic parsing ability and generalization to novel databases through evaluation on various semantic parsing datasets. Second, we test transfer transfer capability to out-of-distribution Table/KG Question Answering tasks to demonstrate induced reasoning capabilities.

**Text-to-SQL Task:** For the Text-to-SQL Task, we use BIRD minidev[6], Spider Yu et al. (2018), EHRSQL Lee et al. (2022) and SM3 sivasubramaniam2024smtexttoquery benchmarks. EHRSQL is specifically chosen to assess generalization to entirely unseen databases; this is ensured by excluding all MIMIC Johnson et al. (2016) database-related questions from our training set since EHRSQL is based on MIMIC. Spider serves as a comprehensive benchmark for text-to-SQL capabilities.

---

[5]https://huggingface.co/datasets/Clinton/Text-to-sql-v1
[6]https://GitHub.com/bird-bench/mini_dev

Table B1: **Overview of Datasets used for Training and Evaluation**. The table shows the datasets used for different stages: Supervised Fine-Tuning (SFT), GRPO Training, and two types of evaluations.

| Stage | Type | Table Task | Graph Task |
|---|---|---|---|
| Training | Semantic Parsing (SFT) | Text-to-SQL Collection* | Neo4j Text-to-Cypher Ozsoy et al. (2024) |
| | Semantic Parsing (GRPO) | BIRD Li et al. (2023) | Neo4j Text-to-Cypher Ozsoy et al. (2024) |
| Evaluation | Semantic Parsing | BIRD minidev[†], EHRSQL Lee et al. (2022), Spider Yu et al. (2018) SM3 Sivasubramaniam et al. (2024) | Neo4j Text-to-Cypher Ozsoy et al. (2024) SM3 Sivasubramaniam et al. (2024) |
| | Structural QA | CRT-QA Zhang et al. (2023), TableBench Wu et al. (2025) | CR-LT-KGQA Guo et al. (2024) |

*`https://huggingface.co/datasets/Clinton/Text-to-sql-v1`

[†]`https://GitHub.com/bird-bench/mini_dev`

**Text-to-Cypher Task:** Evaluation is conducted on the held-out test split of our training data Ozsoy et al. (2024) and on SM3 sivasubramaniam2024smtexttoquery.

**Tabular QA:** To assess transfer to table understanding, we evaluate on CRT QA Zhang et al. (2023) and Tablebench Wu et al. (2025), which focus on direct reasoning over table data.

**KG QA:** We assess knowledge graph reasoning capabilities through evaluation on CR-LT KGQA Guo et al. (2024), which requires direct reasoning over graph structures.

## C    REINFORCEMENT LEARNING DETAILS

We employ GRPO, an RL method originally introduced in Shao et al. (2024). This approach enhances traditional RL by comparing multiple outputs for the same input and assigning relative rewards, enabling more nuanced feedback for complex reasoning tasks.

Formally, for a given natural language question $q$ and its associated database schema, the model generates a set of $G$ candidate SQL or Cypher queries $\{o_1, o_2, \ldots, o_G\}$. Each candidate is evaluated using a task-specific reward function, and the relative advantage $A_i$ is computed for each output. The optimization objective is given by:

$$J_{GRPO}(\Theta) = \mathbb{E}\left[\frac{1}{G}\sum_{i=1}^{G}\min\left(\frac{\pi_\theta(o_i|q)}{\pi_{\theta_{old}}(o_i|q)}A_i, \text{clip}\left(\frac{\pi_\theta(o_i|q)}{\pi_{\theta_{old}}(o_i|q)}, 1-\epsilon, 1+\epsilon\right)A_i\right)\right] - \beta D_{KL}(\pi_\theta||\pi_{\text{ref}})$$

Here, $\pi_\theta$ represents the current policy, $\pi_{\theta_{old}}$ denotes the policy before the update, and $\pi_{\text{ref}}$ is a frozen reference policy used for regularization. The hyperparameters $\epsilon$ and $\beta$ control the clipping threshold and divergence penalty, respectively.

## D    ADDITIONAL RESULTS

Tables D1, D2 show additional results on the Qwen3-14B model.

## E    ABLATION DETAILS

The full ablation results from training Qwen2.5-1.5B instruct are presented in table E1 and E2. For their discussion we refer to Section 4.5.

Table D1: **In-context learning performance on Text-to-SQL (Spider, EHRSQL, BIRD minidev with evidence) and Text-to-Cypher (Text2Cypher) tasks**. Metrics include exact match (EM), execution accuracy (EXE), BLEU score and execution-based F1 score ($F1_{exe}$, per Lee et al. (2022) ). Models are labeled as 'trained-SQL', 'trained-Cypher', or 'trained-Both' based on fine-tuning.

| Model | Text2Cypher | | BIRD | Spider | EHRSQL |
| --- | --- | --- | --- | --- | --- |
| | EM | BLEU | EXE | EXE | $F1_{exe}$ |
| Qwen3-14B | 3.0 ±1.0 | 21.3 ±1.2 | 51.0 ±2.8 | 73.7 ±2.2 | 38.5 ±2.9 |
| Qwen3-14B-trained-SQL | 3.5 ±1.1 | 22.5 ±1.3 | 54.5 ±2.5 | **76.8** ±1.9 | 40.5 ±2.6 |
| Qwen3-14B-trained-Cypher | 5.5 ±1.2 | 26.8 ±1.4 | 52.2 ±2.6 | 74.4 ±1.8 | 39.5 ±2.7 |
| Qwen3-14B-trained-Both | 6.2 ±1.0 | **27.9** ±2.1 | **54.7** ±2.3 | 76.5 ±2.6 | **43.2** ±1.8 |

Table D2: **Out-of-context performance on structural question answering**. Results are reported on tabular (CRT-QA, TableBench) and knowledge graph QA (CR-LT-KGQA). Metrics include exact match (EM) and ROUGE. Models are labeled as 'trained-SQL', 'trained-Cypher', or 'trained-Both' based on fine-tuning. Top two results per dataset are bolded.

| Model | CRT-QA | TableBench | | CR-LT KGQA |
| --- | --- | --- | --- | --- |
| | EM | EM | Rouge | EM |
| Qwen3-14B | 53.3 ±3.6 | 66.3 ±3.5 | 72.1 ±2.7 | 82.8 ±2.0 |
| Qwen3-14B-trained-SQL | 56.2 ±3.2 | 67.8 ±3.7 | 72.8 ±2.9 | 83.7 ±1.9 |
| Qwen3-14B-trained-Cypher | 54.9 ±3.5 | 67.0 ±3.6 | 73.0 ±3.0 | 84.5 ±1.7 |
| Qwen3-14B-trained-Both | 55.6 ±3.1 | 67.0 ±3.8 | 73.0 ±3.1 | 84.2 ±1.8 |

# F  SM3: DIRECT COMPARISON OF SQL, CYPHER, AND MQL

We evaluate models trained on our Text-to-SQL and Text-to-Cypher datasets using the SM3-Text-to-Query Dataset Sivasubramaniam et al. (2024), which provides semantically equivalent queries across SQL, Cypher, and MongoDB Query Language (MQL). This allows us to assess how well our training approach generalizes to unseen query languages. We use the schema information and the few-shot examples from the paper, and use o4 mini as an LLM judge.

Base models demonstrate better performance on SQL compared to Cypher and significantly weaker performance on MQL, suggesting that their behavior reflects the data availability during pretraining (SQL being the most abundant and MQL the scarcest). However, our models outperform in Cypher over SQL, likely due to Cypher's relative simplicity or the greater effectiveness of our structural reward mechanism for graph-based tasks.

Our results in Table F1 demonstrate strong transfer learning capabilities. Models trained jointly on SQL and Cypher (e.g., Qwen2.5-14B-trained-Both achieving 42.5% on Cypher and 40.0% on SQL) outperform their single-task counterparts (Qwen2.5-14B-trained-Cypher achieving 41.8% on Cypher, Qwen2.5-14B-trained-SQL achieving 39.5% on SQL). More importantly, this joint training enables better generalization to the completely unseen MQL, with QwQ-32B-trained-Both reaching 20.8% accuracy compared to base model's 19.4%. Adding few-shot examples further boosts performance across all settings, with QwQ-32B-trained-Both-fewshot achieving our best results (57.7% Cypher, 48.8% SQL, 24.0% MQL), suggesting effective combination of learned structural understanding and exemplar-based reasoning.

Table E1: **SFT vs. SFT+GRPO Ablation**. Comparative performance across training regimes (Base, SFT-only, SFT+GRPO) for SQL-only, Cypher-only, and Joint training. Using Qwen2.5-1.5B-instruct.

|  | Text-to-SQL | | Tabular QA | | Text-to-Cypher | KG QA |
|---|---|---|---|---|---|---|
| **Training Regime** | Spider | EHRSQL | CRT-QA | TableBench | BLEU | CR-LT-KGQA |
| *Base (no SFT/GRPO)* | 24.8 | 4.8 | 5.6 | 12.0 | 4.6 | 26.2 |
| *SFT-only* | 37.1 | 7.1 | 8.4 | 18.0 | 6.9 | 39.2 |
| *SFT+GRPO* | 49.5 | 9.5 | 11.2 | 24.0 | 9.2 | 52.3 |
| *Base (no SFT/GRPO)* | 22.6 | 3.9 | 5.1 | 11.3 | 7.7 | 29.2 |
| *SFT-only* | 33.9 | 5.9 | 7.7 | 16.9 | 11.6 | 43.7 |
| *SFT+GRPO* | 45.2 | 7.8 | 10.2 | 22.5 | 15.4 | 58.3 |
| *Base (no SFT/GRPO)* | 25.6 | 4.7 | 7.1 | 15.0 | 7.9 | 30.4 |
| *SFT-only* | 38.4 | 7.1 | 10.7 | 22.5 | 11.9 | 45.6 |
| *SFT+GRPO* | 51.2 | 9.4 | 14.2 | 30.0 | 15.8 | 60.8 |

Table E2: **Reward Ablation Results**. Performance comparison of the Qwen2.5-1.5B-instruct model trained on Text-to-SQL collection, Neo4j Text-to-Cypher dataset, or both datasets jointly. The three digits in model names indicate the use of different rewards during training[*]. We report execution accuracy for SQL tasks (BIRD minidev + evidence, Spider), Exact Match for QA tasks (CRT-QA, CR-LT-KGQA, TableBench), BLEU score for Text2Cypher, and $f1_{exec}$ for EHRSQL.

|  | Text-to-SQL | | | Tabular QA | | Text-to-Cypher | KG QA |
|---|---|---|---|---|---|---|---|
| **Model** | BIRD minidev | Spider | EHRSQL | CRT-QA | TableBench | Text2Cypher | CR-LT-KGQA |
| SQL-111 | 11.3±1.4 | 49.5±3.1 | **9.5**±2.5 | 11.2±2.1 | 24.0±2.8 | 9.2±1.5 | 52.3±4.6 |
| SQL-011 | 9.5±1.5 | 46.2±3.3 | 8.0±2.6 | 9.7±2.3 | 20.4±3.0 | 7.9±1.4 | 50.2±4.9 |
| SQL-101 | 10.7±1.4 | 47.5±3.2 | 8.7±2.8 | 10.0±2.2 | 22.2±2.9 | 8.7±1.4 | 51.1±4.8 |
| SQL-110 | 10.5±1.6 | 46.7±3.1 | 8.2±2.4 | 9.0±2.1 | 19.3±2.7 | 8.3±1.5 | 49.5±4.5 |
| Cypher-111 | 8.2±2.3 | 45.2±4.2 | 7.8±2.5 | 10.2±2.5 | 22.5±3.1 | 15.4±1.2 | 58.3±3.9 |
| Cypher-011 | 7.5±2.4 | 45.1±4.0 | 7.5±2.6 | 9.8±2.3 | 20.1±3.4 | 13.2±1.4 | 54.7±3.8 |
| Cypher-101 | 7.7±2.5 | 45.5±4.5 | 7.7±2.7 | 9.6±2.4 | 20.6±3.0 | 14.5±1.9 | 56.3±4.2 |
| Cypher-110 | 7.6±2.3 | 45.3±4.4 | 7.6±2.8 | 9.3±2.4 | 19.7±3.2 | 13.8±1.2 | 54.2±3.7 |
| Both-111 | **12.0**±1.2 | **51.2**±3.3 | 9.4±2.4 | **14.2**±2.8 | **30.0**±3.1 | **15.8**±2.2 | **60.8**±4.0 |
| Both-011 | 9.8±1.7 | 48.1±3.4 | 9.0±2.6 | 12.0±2.3 | 26.5±3.2 | 12.5±1.3 | 57.0±4.3 |
| Both-101 | 9.5±2.3 | 49.3±3.5 | 9.4±2.3 | 13.0±2.3 | 27.3±3.0 | 13.8±1.4 | 58.5±4.1 |
| Both-110 | 9.3±2.2 | 48.9±3.6 | 9.1±2.5 | 11.5±2.4 | 25.8±3.5 | 13.2±1.4 | 56.4±4.2 |
| Qwen2.5-1.5B-instruct | 9.0±1.6 | 44.7±3.6 | 7.4±3.2 | 8.3±2.3 | 18.0±2.4 | 7.2±1.3 | 48.7±5.5 |

[*]Model names (e.g., SQL-111) use a three-digit format where each digit (1/0) indicates whether the corresponding reward was used during training: string matching (1st digit), LLM reward (2nd digit), and structural reward (3rd digit).

# G  PROMPTS

This section details the prompts used in our pipeline, closely aligned with Stoisser et al. (2025). For Cypher queries, we simply replace the word "SQL" with "Cypher" in all prompts.

## G.1  INSTRUCTION SUMMARIZATION

As part of the synthetic data generation pipeline, we employ a specialized prompt to distill complex natural language SQL task descriptions into concise, structured summaries that capture essential query requirements.

> You are a SQL assistant tasked with summarizing instructions for SQL query generation. Below are original natural language task descriptions that outline a specific request. Create a concise summary that highlights the core requirements for the SQL query to be constructed.
> **ORIGINAL INSTRUCTIONS: original natural language task description**
> When creating the summary, focus on clarity and precision, ensuring that the essential elements necessary for generating the SQL query are retained. You may follow this template:
> **SUMMARY:** Write the summarized instructions here, clearly stating the goals and key aspects required for the SQL query.

Table F1: **SM3 - Direct Comparison of SQL, Cypher, and MQL.** Performance on semantically equivalent queries across three query languages using the SM3 benchmark. Each query expresses the same intent in SQL, Cypher, and MongoDB Query Language (MQL), enabling direct comparison of language complexity. LLM execution accuracy is reported. Models fine-tuned on Text-to-SQL are labeled as 'trained-SQL', those fine-tuned on Text-to-Cypher as 'trained-Cypher', and those trained on both tasks as 'trained-Both'. Few-shot results indicate performance when correct Text-to-Query examples are provided in the prompt.

| Model | Text-to-Cypher | | Text-to-SQL | | Text-to-MQL | |
| --- | --- | --- | --- | --- | --- | --- |
| | Zero-shot | Few-shot | Zero-shot | Few-shot | Zero-shot | Few-shot |
| o3 | 40.5±3.9 | **59.3±3.8** | **46.3±3.2** | 57.7±3.6 | **25.7±4.1** | **30.3±3.5** |
| o3-mini | 40.0±5.6 | 53.7±3.3 | 41.3±4.2 | 50.8±3.7 | 15.5±2.5 | 20.0±2.1 |
| Qwen2.5-14B-instruct | 32.5±3.2 | 36.7±4.2 | 35.5±3.7 | 41.7±3.8 | 9.0±3.9 | 15.2±2.2 |
| Qwen2.5-14B-trained-SQL | 35.5±2.8 | 38.5±3.8 | 39.5±4.0 | 43.7±3.3 | 9.6±2.2 | 18.2±1.8 |
| Qwen2.5-14B-trained-Cypher | 41.8±3.3 | 49.8±3.9 | 36.7±5.3 | 40.5±4.0 | 10.3±1.2 | 18.5±1.6 |
| Qwen2.5-14B-trained-Both | 42.5±3.2 | 49.9±4.3 | 40.0±4.2 | 44.3±4.3 | 13.8±1.1 | 19.7±2.4 |
| Qwen3-14B | 40.5±5.6 | 53.0±4.0 | 38.3±4.8 | 50.7±4.1 | 19.7±2.0 | 19.0±3.5 |
| Qwen3-14B-trained-SQL | 40.7±4.2 | 53.5±3.9 | 41.8±4.0 | **51.8±3.3** | 20.0±2.1 | 20.7±1.7 |
| Qwen3-14B-trained-Cypher | 45.2±3.8 | 54.0±4.1 | 40.5±4.7 | 47.5±3.5 | 20.5±2.0 | 20.5±1.9 |
| Qwen3-14B-trained-Both | **45.7±5.6** | 54.0±4.1 | 42.0±2.2 | 48.0±3.8 | 20.5±2.8 | 20.6±0.4 |
| QwQ-32B | 35.8±5.2 | 55.3±4.3 | 41.0±4.2 | 42.4±4.0 | 19.4±2.4 | 20.8±2.7 |
| QwQ-32B-trained-Both | 43.7±5.5 | **57.7±4.3** | **42.5±4.7** | 48.8±3.8 | **20.8±1.2** | **24.0±2.3** |

## G.2 Creating Synthetic CoT

Our synthetic Chain-of-Thought generation prompt combines schema information, original instructions, and summarized requirements to guide the creation of well-reasoned SQL queries.

> You are a SQL expert. Below are SQL table schemas paired with both original and summarized instructions that describe a specific task. Using valid SQLite syntax, write a response that appropriately completes the request for the provided tables.
> **SCHEMA: schema**
> **ORIGINAL INSTRUCTIONS: original natural language task description**
> **SUMMARIZED INSTRUCTIONS: summarized task instructions**
> When answering, provide reasoning for the SQL query you create using the following template:
> <sql> Write the SQL query here, ensuring it adheres to SQLite syntax and effectively accomplishes the task described in the instructions. </sql>

## G.3 Evaluation of Synthetic CoT

The evaluation prompt for the synthetic CoT generation pipeline provides a binary assessment framework for comparing generated SQL queries against reference solutions while considering schema constraints.

> You are an SQL expert, and your task is to evaluate whether the SQL query below is correct based on the provided schema and the correct SQL reference.
> **SQL Query:** ans.sql
> **Schema:** schema
> **Correct SQL:** correct_sql
> Return ONLY "Correct" or "Wrong".

## G.4 TRAINING PROMPTS

Following Guo et al. (2025), we use this system prompt that encourages explicit reasoning through Chain-of-Thought.

> A conversation between User and Assistant. The user asks a question, and the Assistant solves it, The assistant first thinks about the reasoning process in the mind and then provides the user with the answer. The reasoning process is enclosed within <think> </think> tags, respectively, i.e., <think> reasoning process here </think> answer here. User:

The user prompt provides specific instructions for query generation:

> Below is an instruction that describes a task, paired with an input that provides further context. Write a response that appropriately completes the request.
> **Instruction:** You are a SQLite expert. Given an input question, create a syntactically correct SQLite query to run. Enclose the final sql query within "'''sql" and "'''" tags.
> **Input:** Here is the relevant table info: {table_info}.
> Write a SQLite query for the following task: {task}.
> **Response:**

## G.5 LLM EVALUATION METRIC PROMPT

For binary evaluation of query correctness, we employ a prompt that focuses on semantic equivalence while allowing for minor syntactic variations:

> You are SQL expert and your task is to evaluate if the predicted SQL query is correct based on the Schema and the correct SQL query. If no SQL query was found then the answer is Wrong. The query is considered correct even if the only mistakes are in letter casing (uppercase vs lowercase).
> **Schema:** {example['context']}
> **Predicted query:** {pred_query}
> **Correct SQL query:** {correct_query}
> Return ONLY "Correct" or "Wrong"

## G.6 LLM JUDGE REWARD CLASSIFICATION PROMPT

To provide fine-grained feedback during training, we use a five-class classification prompt that assesses query quality across multiple dimensions:

> Compare these SQL queries to the correct query and grade each one as: 'Very bad', 'Bad', 'Above average', 'Good', or 'Excellent'.
> Use the following grading system, with the correct query as reference:
> **Correct Query:** {true_query}
> **1. Excellent:** Perfect match with {true_query}
> **2. Good:** Contains only grammar mistakes
> **3. Above average:** Mostly correct but contains one logical error
> **4. Bad:** Contains multiple mistakes
> **5. Very bad:** No query produced or significantly different from correct query
> **Queries to grade:** {queries_to_rank}
> {format_instructions}

