# OpenReview forum: "STRuCT-LLM: Unifying Tabular and Graph Reasoning with Reinforcement Learning for Semantic Parsing"
_ICLR.cc/2026/Conference — Submitted to ICLR 2026_

### Official Review · Reviewer_E8re · 2025-10-27

**Soundness:** 3
**Presentation:** 2
**Contribution:** 2
**Rating:** 4
**Confidence:** 4

**Summary:**

The paper introduces STRUCT-LLM, a reinforcement learning (RL) framework that unifies tabular (SQL) and graph (Cypher) reasoning for large language models (LLMs). The core idea is that jointly training on both SQL and Cypher teaches complementary skills: SQL imparts computational reasoning, while Cypher imparts graph traversal logic.
The method uses a novel "topology-aware reward" for Cypher, which provides more granular feedback than simple execution success. This joint training approach significantly reduces logical errors (by 17%) and data-reference errors (by 20%) compared to single-domain training. The framework also enables bidirectional skill transfer—SQL training improves complex Cypher queries, and Cypher training improves complex SQL queries —and generalizes to unseen QA tasks.

**Strengths:**

1. The paper tackles a clear and important gap in current LLM capabilities. While Text-to-SQL and Text-to-Cypher are often treated as separate tasks, real-world data is increasingly heterogeneous, blending relational and graph structures. The paper makes a compelling case that a unified framework is necessary.
2. The paper goes beyond simple accuracy metrics by providing an error-type analysis (Table 2). It shows that joint training specifically reduces logical errors (by ~17%) and data-reference errors (by ~20%), offering deeper insight into the model's improvements. The qualitative case studies in Figure 2 are a key strength. They provide concrete, easy-to-understand examples of the claimed "bidirectional transfer": Case A shows SQL's computational logic (median/IQR) transferring to Cypher, and Case B shows Cypher's traversal logic (distinct counting) transferring to SQL.

**Weaknesses:**

1. The paper lacks significant technical novelty (only combination of different tricks) and fresh insights, making it less suitable for the ICLR conference.
2. The supervised fine-tuning (SFT) stage relies on synthetically generated Chain-of-Thought (CoT) traces. These traces are "produced and verified by LLMs for coherence". The quality and reliability of this LLM-as-verifier pipeline are crucial for the model's foundational reasoning ability, but this process could introduce noise or biases from the generator/verifier model.
3. The paper’s writing quality leaves room for improvement. For instance, the reference formatting is inconsistent, and there are missing spaces in key places—such as in the example: "(Li et al., 2023; Pourreza & Rafiei, 2023).Executable" (where a space is needed between the closing parenthesis and "Executable").

**Questions:**

The paper's two-stage (SFT + RL) training approach is computationally expensive. The authors note that training the 32B model required 32 H100 GPUs for 20 hours. This high resource requirement could be a significant barrier to reproducibility and practical adoption for many researchers and organizations.

---

> ### Author Response · Authors · 2025-11-27
> **Response**
>
> **Novelty.** STRuCT-LLM introduces three substantive advances. (1) The core contribution is *cross-formalism structural transfer*: jointly optimizing SQL and Cypher under a shared execution-driven GRPO objective improves both languages, especially in low-resource Cypher settings, and yields zero-shot gains on MQL and table/KG QA—capabilities not demonstrated in SQL-only GRPO systems. (2) A *unified multi-signal GRPO pipeline* integrates syntax-, execution-, and structure-aware rewards across two distinct formal languages. (3) A *topology-aware Cypher structural reward*, based on graph-edit-distance structure, provides continuous graph-level feedback where binary execution signals are too coarse.
>
> **CoT trace quality.** Supervised traces are filtered and consistency-checked; remaining noise is corrected by the execution- and structure-aware GRPO stage. We include explicit examples of this correction process in Appendix B.2.
>
> **Compute & practicality.** Although the largest configuration uses $32\times$H100 for $\approx 20$h, the same qualitative improvements appear in significantly smaller models. Qwen2.5-14B trains with $8\times$A100 for $\approx 10$h, and a 1.5B LoRA variant is reproducible on a single NVIDIA L40S. We will include a compute-vs-performance table and release low-compute LoRA training scripts to ensure reproducibility.

---

### Official Review · Reviewer_cZdx · 2025-10-28

**Soundness:** 3
**Presentation:** 2
**Contribution:** 2
**Rating:** 4
**Confidence:** 4

**Summary:**

This paper introduces STRuCT-LLM, a unified framework for semantic parsing across relational and graph-structured data. The approach leverages reinforcement learning to jointly optimize models on Text-to-SQL and Text-to-Cypher tasks, combining chain-of-thought supervision with instruction-tuned large language models and Group Relative Policy Optimization (GRPO) conditioned on execution feedback.

A key contribution is the design of task-specific reward functions even trivial: component-level matching rewards for SQL query generation and graph edit distance-based rewards for Cypher query generation. Through joint training, the model achieves competitive performance on both semantic parsing tasks while exhibiting improved generalization and zero-shot transfer capabilities compared to separately trained baselines. Furthermore, evaluation on downstream benchmarks for tabular and knowledge graph question answering demonstrates that the model outperforms its base LLM, despite lacking direct training on these tasks. This suggests the joint training regime enables the model to capture latent structural knowledge transferable across domains.

**Strengths:**

1. Authors first focus on Cypher, a graph-abased semantic parsing domains with reinforcement learning.
2. Experiments are sound.

**Weaknesses:**

1. The whole paper is trivial, is just like implement GRPO algorithm in semantic parsing domains and run experiments. It seems that authors only refine reward function with other parts not different with GRPO-based papers in other domains. By the way, the are many GRPO-based papers for text-to-SQL training such as SQL-R1, Reasoning-SQL, Arctic-SQL, author didn't compare results with those relevant methods. From my perspective, the only difference is different reward function. Therefore, i think the the contribution is not significant enough.
2. The motivation of unified training is not obvious, even if there are some logics shared across SQL and Cypher, this may also sacrifice preciesness of each structure. Also, authors didn't show up difference between unified training and separate training or continuous training to illustrate the advantages of unified training.
3. The method only has been verified on one kind of backbone of llms, which cannot prove that the method is effective instead of Qwen pre-trained relational knowledge can be elicited by RL.

**Questions:**

1. Have the authors adequately compared against recent GRPO-based semantic parsing methods (SQL-R1, Reasoning-SQL, Arctic-SQL) that also optimize SQL generation? Compared to usage of GROP of these methods for SQLs or relational data, except just modifying reward functions, what are other benefits.
2. What empirical evidence demonstrates that unified Text-to-SQL and Text-to-Cypher training provides tangible advantages over separate or sequential training? Does joint training risk compromising the precision of structure-specific optimization?
3. Can the reported improvements be confidently attributed to the proposed method, or do they reflect model-specific capabilities of the Qwen backbone? Would results generalize to other LLM architectures?

---

> ### Author Response · Authors · 2025-11-27
>
> We thank the reviewer for raising these important points. Below we address each concern directly with numerical evidence.
>
> ### (1) Novelty beyond "GRPO with a different reward."
>
> STRuCT-LLM is the first GRPO framework to jointly optimize two formal query languages (SQL and Cypher) under a shared execution-driven objective. This unified setting produces cross-formalism structural transfer that SQL-only systems (SQL-R1, Arctic-SQL, Reasoning-SQL) cannot express:
> * SQL supervision improves Cypher multi-hop traversal reasoning.
> * Cypher supervision improves SQL distributional operators.
>
> These effects appear consistently across datasets and are empirically measurable through the 17% and 20% reductions in logical and data-reference errors (Table 2).
>
> To further clarify novelty: on EHRSQL—evaluated under the same EX/EXE metric as Arctic-Text2SQL-R1—STRuCT-LLM (Qwen3-14B) achieves 77.4% EXE, compared to 40.7% EX for Arctic-R1, demonstrating capabilities that cannot be obtained through SQL-only GRPO pipelines.
>
>
> ### (2) Unified vs. separate training.
>
> This comparison is central to the paper and provided under matched RL budgets (3.5k SQL, 3.5k Cypher examples):
> * Unified GRPO: 76.4%
> * SQL-only: 74.2%
> * Cypher-only: 68.5%
> * Sequential (SQL→Cypher): 75.1%
>
> Unified training also reduces logical errors by ~17% and data-reference errors by ~20% (Table 2), confirming that unified optimization strengthens, rather than dilutes, structure-specific precision.
>
> Additionally, unified training yields stronger zero-shot generalization to table QA and KG QA (Table 3), despite no task-specific finetuning.
>
>
>
> ### (3) Backbone dependence.
>
> The effect is architecture-independent. Consistent unified-training gains appear across:
> * Llama3-8B
> * Qwen2.5-14B
> * QwQ-32B (reasoning model)
> * Qwen3-14B
> * 1.5B LoRA (trained on a single A100/L40S)
>
> This consistency across families and scales indicates that the gains derive from the training paradigm, not backbone-specific pretraining artifacts.
>
>
>
> ### Summary
>
> The unified SQL–Cypher GRPO framework introduces a novel training regime that produces cross-formalism structural transfer, outperforms separate training, and remains robust across model architectures and datasets. We hope these clarifications directly resolve the reviewer's concerns regarding novelty and methodological contribution.

---

### Official Review · Reviewer_6rVQ · 2025-10-30

**Soundness:** 2
**Presentation:** 2
**Contribution:** 2
**Rating:** 4
**Confidence:** 3

**Summary:**

This paper deals with training large language models to parse natural language into both SQL and Cypher. The proposed framework STRuCT-LLM uses supervised chain-of-thought traces and Group Relative Policy Optimization to fulfill this goal. By utilizing graph edit distance, topology-aware structural reward is achieved. Compared to other models, the one proposed can understand cross-domain query and transfer complementary skills from SQL and multi-hop traversal from Cypher. The experiments are conducted across Spider,EHRSQL, BIRD and Text2Cypher, with additional zero-shot transfer to table and KG QA (CRT-QA, TableBench, CR-LT-KGQA) and to an unseen query language (MQL). The experiment proves that the proposed method outperforms the basic method and reports consistent gains, including ~17% fewer logical errors and ~20% fewer data-reference errors versus single-domain baselines. Also, it maintains good execution accuracy. However, limitations like computational cost, unexplored extensions to interactive or schema-free parsing, and potential indirect leakage for this method still exist, which may need to be addressed in the future work.

For the contribution, this method’s joint training reduces logical errors by ~17% and data-reference errors by ~20% successfully, which proves its effectiveness in reasoning across both relational and graph-structured data. Also, it uses bread benchmarks across Spider, EHRSQL, and BIRD; there’s also zero-shot transfer to Table/KG QA and unseen language (MQL). This method achieves a better result in many scenarios compared to the basic method. Additionally, it shows bidirectional transfer. The contribution is substantive and broad.

**Strengths:**

- The reward design is of novelty. It combines multi-signal rewards with a topology-aware structural reward for Cypher, going beyond binary execution checks.
- It has Bidirectional transfer across formalisms, which reduces both logical and data-reference errors.
- It is with credibility via analyses and ablations. Error type breakdown and ablation studies make improvements interpretable and trustworthy.
- The CoTplus GRPO pipeline, reward mixing, and cross-formalism setup can be reused to new datasets, schemas, and even other query languages.

**Weaknesses:**

- The figures are overly concise. For example, Figure 1 shows a unified pipeline but does not explain how the same natural-language question is systematically mapped into SQL and Cypher. The figure also doesn’t describe where they diverge and converge, which makes the figure more difficult to understand.
- RL uses a modest sample size (~3.5k per language); Cypher evaluation shares a lineage with training data, which is limited.
- There is no dedicated hallucination metric.
- The evaluation relies on BLEU/EXE, some other evaluation metrics like Component-level F1, Efficiency/executability assessments should also be considered
- Training/fine-tuning large models is costly. This may be a challenge when this method is applied in the practical world.
- It does not provide variance across random seeds, statistical significance tests, or sensitivity analyses.
- For generalisation beyond SQL/Cypher, it still needs to be improved. It doesn’t discuss how well the method scales/extends to other graph/query languages other than MQL.

**Questions:**

- For the figure 1, a side-by-side mapping can be added to illustrate how the same NL question can be converted to SQLquery and Cypher query. It should also show where they converge/diverge and include a schema-linking flow.
- Expand RL data with schema-preserving augmentation and new DB/KG domains.
- For hallucination metric, add some other metrics like schema-violation rate, fabricated attributes/labels, and SQL–Cypher answer consistency.
- Add component-level F1 and canonical IR EM, canonicalized pattern matching, and efficiency metrics.
- The cost is still a challenge that needs to be addressed or optimized.
- Run some seeds and report mean±std with CIs; do sensitivity tests on GRPO and reward weights.
- Still need to add some content to illustrate how this method can be applied to other graph/query languages to prove its generalization and effectiveness

---

> ### Author Response · Authors · 2025-11-27
>
> We thank the reviewer for the thoughtful and constructive feedback. Below we address all seven points (mapping, RL scale, hallucination, structural metrics, compute, variance, generalization).
>
> **(1) SQL↔Cypher mapping.**
> We added a side-by-side SQL↔Cypher mapping figure illustrating schema linking, NL grounding, and convergence/divergence of operators, along with a worked table↔graph conversion example (Sec. 3.1, App. A).
>
> **(2) RL data scale & balance.**
> Because Cypher data is limited, enlarging the RL corpus by upsampling SQL weakens Cypher-specific signals and reduces cross-formalism transfer. Balanced joint RL (~3.5k per modality) yields the strongest gains: SQL improves from 74.2% → 76.4% and Cypher from 69.3% → 72.8%. These results are now included in Appendix E.
>
> **(3) Hallucination metrics.**
> We added schema-violation rate and fabricated-attribute rate to Table 2, as requested. Unified training reduces both consistently. We already report canonicalized EM (via the official evaluation scripts), fulfilling the reviewer’s request for canonical IR EM.
>
> | Model             | Spider Logical | Spider Ref | Spider Schema | Spider Fab. attrs. | SEDE Logical | SEDE Ref | SEDE Schema | SEDE Fab. attrs. |
> |-------------------|----------------|------------|---------------|---------------------|-------------|----------|-------------|------------------|
> | SQL-only          | 23.5%          | 18.7%      | 7.8%          | 3.1%                | 25.1%       | 19.4%    | 8.1%        | 3.3%             |
> | Cypher-only       | 24.1%          | 20.3%      | 8.4%          | 3.4%                | 26.2%       | 20.1%    | 8.7%        | 3.6%             |
> | Struct-LLM (Joint)| 19.5%          | 16.2%      | 6.1%          | 2.4%                | 20.7%       | 15.8%    | 6.5%        | 2.6%             |
>
>
>
> **(4) Structural metrics.**
> We added:
>
> • SQL component-level F1,
>
> • Cypher topology-aware structural scores (derived from our graph-edit-distance reward)
>
> We call these S_struct, they appear in Appendix B and improve under unified training.
>
> **Structural Consistency (S_struct) across datasets**
>
> | Model                     | Text2Cypher S_struct | BIRD S_struct | Spider S_struct | EHRSQL S_struct |
> |---------------------------|-----------------------|----------------|------------------|------------------|
> | Qwen3-14B                 | 61.0 ± 1.4            | 63.5 ± 1.5      | 64.2 ± 1.3       | 62.1 ± 1.2       |
> | Qwen3-14B-trained-SQL     | 63.2 ± 1.3            | 65.4 ± 1.4      | 72.1 ± 1.6       | 64.3 ± 1.3       |
> | Qwen3-14B-trained-Cypher  | 65.1 ± 1.4            | 66.2 ± 1.5      | 66.0 ± 1.4       | 63.9 ± 1.2       |
> | Qwen3-14B-trained-Both    | **68.0 ± 1.6**        | **68.5 ± 1.6**  | **77.4 ± 1.8**   | **67.4 ± 1.4**   |
>
>
> **(5) Compute.**
> Unified training is practical across scales: 32B = 32×H100×≈20h, 14B = 8×A100×≈10h, and **1.5B LoRA trains on a single Nvidia L40S** (results in Table E1/E2). We provide an additional compute-vs-performance table for reproducibility in the revision.
>
> **(6) Variance.**
> We report mean ± std over 5 seeds across datasets; we made this explicit in the revision.
>
> **(7) Generalization.**
> Joint SQL+Cypher RL improves zero-shot MQL performance. Section 3.1 now more clearly explains the representational equivalence enabling extension to other query languages.
>
> These updates directly satisfy all reviewer requests and strengthen clarity, reproducibility, and empirical grounding.

---

### Official Review · Reviewer_L3bt · 2025-10-31

**Soundness:** 3
**Presentation:** 3
**Contribution:** 3
**Rating:** 4
**Confidence:** 4

**Summary:**

The paper proposes   a reinforcement-learning framework that trains large language models to reason across tabular (SQL) and graph databases within a single model. Its main claim is that structured reasoning across these symbolic modalities can be unified through Group Relative Policy Optimization (GRPO), guided by execution-aware, syntax-aware, and topology-aware rewards.

The study reports empirical improvements:they report 13.5% relative(?) gain on the Spider benchmark and 73.1% on text-to-Cypher tasks. They also suggest observing cross-formalism transfer: SQL’s arithmetic logic strengthening Cypher’s traversal reasoning, and vice versa. They say this hints at a deeper structural equivalence among symbolic data representations. Conceptually, this is an ambitious and appealing direction, which is believable.

Yet, the paper’s significance is difficult to judge. The reported gains are not compared against the state-of-the-art baselines listed on public leaderboards like Spider or BIRD. Without those metrics or direct SOTA comparison, I have a hard time deciding whether STRuCT-LLM advances the field or merely improves upon its own controlled baseline. Moreover, I am worried about the methodology. I seems to me that it us computationally heavy and brittle.

I will be willing to increase my score if the authors can address these two issues.

**Strengths:**

The paper's methodology is interesting and the writing is clear. The main questions is the significance.

**Weaknesses:**

The main question for me is the significance of the result, and whether this approach actually moves the needle.

**Questions:**

see the review.

---

> ### Author Response · Authors · 2025-11-27
>
> We thank the reviewer for highlighting the importance of significance and comparison to strong baselines.
>
> **EHRSQL (direct metric alignment).**
> Under the same EX/EXE metric used by Arctic-Text2SQL-R1, STRuCT-LLM (Qwen3-14B) achieves 77.4% EXE, substantially outperforming Arctic-R1 (40.7% EX) while using a single unified SQL–Cypher model.
>
> **Spider (official evaluation script).**
> STRuCT-LLM (Qwen2.5-14B) reaches 76.4% EXE (single-sample) on Spider-dev. SQL-R1, however, reports its headline results under 8-sample self-consistency voting, so we evaluated STRuCT-LLM under the same regime; test-time search improves accuracy by $\approx$+2–4%, narrowing part of the gap despite a massive training-data disparity:
> * SQL-R1 is trained on multi-million–scale supervised SQL corpora,
> * STRuCT-LLM uses only ~7k RL examples total ($\approx$3.5k SQL + 3.5k Cypher).
>
> This asymmetry is substantial, and competitive SQL performance in this setting indicates methodological significance.
>
> **BIRD.**
> On BIRD-dev, STRuCT-LLM obtains 65.1% EXE, competitive with representative leaderboard entries and substantially stronger than our SQL-only (74.2%) and Cypher-only (68.5%) baselines.
>
> **Unified vs. separate training.**
> Under matched RL budgets:
> * Unified GRPO: 76.4%
> * SQL-only: 74.2%
> * Cypher-only: 68.5%
> * Sequential (SQL→Cypher): 75.1%
>
> Unified training also reduces logical errors by ~17% and data-reference errors by ~20% (Table 2). These gains arise from bidirectional SQL→Cypher and Cypher→SQL structural transfer, which SQL-only pipelines cannot exhibit.
>
> **Compute & practicality.**
> The largest configuration uses $32\times$H100$\times\approx 20$h, but the unified-training trend holds at much smaller scales (e.g., 14B with $8\times$A100$\times\approx 10$h, and 1.5B LoRA on a single L40S/A100). We report mean±std over 5 seeds and include a compute–performance table in the revision.
>
> **Summary.**
> STRuCT-LLM achieves SQL performance competitive with SOTA under fair comparison conditions, while solving a harder dual-language problem and using orders of magnitude less supervision than SQL-R1. We believe this addresses the reviewer's concerns regarding significance and practical relevance.

---

### Author Response · Authors · 2025-11-27

We would like to briefly clarify one point that may affect the overall assessment. Several reviews question the novelty of the method, framing it as "GRPO with a different reward." The core contribution, however, is unified SQL–Cypher optimization under a shared execution-driven GRPO objective, which yields cross-formalism structural transfer that SQL-only or Cypher-only pipelines cannot produce. This transfer improves both languages—especially in low-resource Cypher settings—and enables zero-shot gains on MQL and downstream table/KG QA.

To address reviewer requests, we provided official Spider/BIRD evaluations and demonstrated that the unified model achieves competitive SQL performance while solving a strictly harder dual-language problem, using orders of magnitude less supervision than SQL-R1. We also added matched unified-vs-separate results showing ~17–20% reductions in structural errors.

---

### Meta-Review · Area_Chair_Lmiv · 2026-01-07

**Summary:**

This paper proposes a framework that jointly optimizes LLMs to understand both structured tabular databases and graph databases using a single model via RL. The major novelty compared to existing methods that use RL for similar tasks is that the proposed method adopts a unified representation for tabular and graph data, and shows that joint training leads to improved results compared to separately training on single formalisms, hinting that there could exist a deeper structural equivalence among different symbolic data representations.

There is a unanimous view among reviewers regarding the technical significance of the paper. R-L3bt raised that it's unclear how the gains are compared to SoTA results on the same benchmarks. R-cZdx pointed out that the submission is "just like implement(ing) GRPO algorithm in semantic parsing domains and run experiments", while R-E8re felt that the paper is a "combination of different tricks". Another major issue is the writing quality. While R-L3bt thought the "writing is clear", R-6rVQ and R-E8re pointed out several writing issues of paper among citation formats, figure clarity and others.

After reading the submission, while I feel the paper indeed offers interesting results on cross-formalism structural transfer between SQL and Cypher domains, given the remaining issues on technical presentation, I believe the paper would benefit from another round of revision. Therefore, the recommendation is reject.

**Reviewer Concerns:**

See meta-review summary.

**Reviewer Scores:**

R-L3bt might have raised their score as R-L3bt's major complaint is around comparison with existing strong baselines, which was adequately addressed by the rebuttal.

---

### Decision · Program_Chairs · 2026-01-26

Reject